# Reconstruction Set Test (RESET): A computationally efficient method for single sample gene set testing based on randomized reduced rank reconstruction error

**H. Robert Frost** *

Department of Biomedical Data Science, Geisel School of Medicine, Dartmouth College, Hanover, New Hampshire, United States of America

* rob.frost@dartmouth.edu

## Abstract

We have developed a new, and analytically novel, single sample gene set testing method called Reconstruction Set Test (RESET). RESET quantifies gene set importance based on the ability of set genes to reconstruct values for all measured genes. RESET is realized using a computationally efficient randomized reduced rank reconstruction algorithm (available via the RESET R package on CRAN) that can effectively detect patterns of differential abundance and differential correlation for self-contained and competitive scenarios. As demonstrated using real and simulated scRNA-seq data, RESET provides superior performance at a lower computational cost relative to other single sample approaches.

**Data Availability Statement:** -An implementation of the RESET method and associated vignettes are available via the RESET R package on CRAN (https://cran.r-project.org/web/packages/RESET/index.html). -The R logic used to generate the

## Author summary

Gene set testing methods are widely used to analyze transcriptomic data with techniques that provide sample level scores increasingly popular given their significant analytical flexibility. For the analysis of single cell data, however, current cell-level methods have several important limitations: poor computational performance, low sensitivity to patterns of differential correlation, and limited support for competitive scenarios that compare set and non-set genes. To address these challenges, we have developed the RESET (Reconstruction Set Test) method that generates overall and cell-level gene set scores using randomized reduced rank reconstruction error. Relative to existing single sample techniques, RESET can more effectively detect patterns of differential abundance and differential correlation under both self-contained and competitive scenarios at a substantially lower computational cost.

## Introduction

### Gene set testing

High-dimensional genomic profiling technologies, such as RNA-sequencing, give researchers a powerful, molecular-level picture of tissue and cellular biology, however, the improvement

simulated results is available at the paper website at https://hrfrost.host.dartmouth.edu/RESET/RESET_simulation_logic.zip. -The gene sets used in the analysis are publicly available from the Molecular Signatures Database (MSigDB) (https://www.gsea-msigdb.org/gsea/msigdb/index.jsp) -The PBMC scRNA-seq data used to generate the results is also used in the Seurat Guided Clustering Tutorial (https://satijalab.org/seurat/articles/pbmc3k_tutorial.html) and freely accessible from 10x Genomics via a Creative Commons Attribution license. A copy of the compressed data file is accessible at https://hrfrost.host.dartmouth.edu/RESET/pbmc3k_filtered_gene_bc_matrices.tar.gz or via the 10x site at https://cf.10xgenomics.com/samples/cell/pbmc3k/pbmc3k_filtered_gene_bc_matrices.tar.gz. This dataset is also available via the SeuratData R package as the pbmc3k dataset. Note that data access via the 10x website requires specification of contact information for marketing purposes. -The mouse brain scRNA-seq data used to generate the results is also freely accessible from 10x Genomics via a Creative Commons Attribution license. A copy of the compressed data file is accessible at https://hrfrost.host.dartmouth.edu/RESET/neuron_10k_v3_filtered_feature_bc_matrix.tar.gz or via the 10x site at https://cf.10xgenomics.com/samples/cell-exp/3.0.0/neuron_10k_v3/neuron_10k_v3_filtered_feature_bc_matrix.tar.gz. Note that data access via the 10x website requires specification of contact information for marketing purposes. -The human cord blood scRNA-seq data used to generate the results is available via the SeuratData R package as the cbmc dataset (https://github.com/satijalab/seurat-data).

**Funding:** Work on this paper by HRF was funded by National Institutes of Health grants R35GM146586, R21CA253408, P20GM130454 and P30CA023108. The funders had no role in the design of the study, collection, analysis, and interpretation of data, or in writing the manuscript.

**Competing interests:** The authors have declared that no competing interests exist.

in fidelity obtained by measuring thousands of genomic variables comes at the price of impaired interpretation, loss of power due to multiple hypothesis correction and poor reproducibility [1, 2]. To address these challenges for bulk tissue data, researchers developed gene set testing, or pathway analysis, methods [3–6]. Gene set testing is a widely used and effective hypothesis aggregation technique that analyzes biologically meaningful groups of genes, e.g., the genes involved in specific signaling pathway defined in a resource like Reactome [7], instead of individual genomic variables. Focusing on a collection of gene sets can significantly improve power, interpretation and replication relative to an analysis focused on individual genes [3, 8]. The benefits of gene set testing are even more pronounced for single cell transcriptomic data given increased technical variance and sparsity [9, 10]. Gene set testing methods can be grouped according to four main features:

1. *Supervised vs unsupervised*: Does the method test for the association between gene set members and a specific outcome or does it generate gene set scores using only the measured genomic data?

2. *Population vs single sample*: Does the method generate gene set scores for each sample or just a single score for the entire population?

3. *Self-contained vs competitive*: Does the method test the $H_0$ that none of the genes in the set has an association with the outcome or the $H_0$ that the genes in the set are not more associated with the outcome than genes not in the set? In other words, self-contained methods only leverage the data associated with genes in the set whereas competitive techniques use all of the measured expression data.

4. *Uniset vs multiset*: Does the method test each gene set separately (uniset) or jointly evaluate all sets in a collection (multiset)?

The most popular type of gene set test is uniset, population-based, competitive and supervised (e.g., GSEA [2] and CAMERA [11]), which is driven by several factors: 1) uniset tests are easier to implement and execute than multiset tests, 2) biological hypotheses of interest typically correspond to supervised tests (e.g., differential expression relative to a specific clinical variable), and 3) a competitive $H_0$ often generates more meaningful results than a self-contained $H_0$ [8]. Although gene set analysis can be performed on a variety of omics data types, it is most commonly applied to transcriptomics data and, without loss of generality, we will assume this data type in the remainder of the manuscript.

## Single sample gene set testing

Although supervised and population-level methods are the most commonly used gene set testing techniques, unsupervised and single sample methods have become increasingly popular given their significant analytical flexibility. Single sample methods, which are inherently unsupervised, operate like a variable transformation to convert an input $n \times p$ matrix $\mathbf{X}$ that captures expression of $p$ genes in $n$ samples into an $n \times m$ matrix $\mathbf{S}$ that captures the sample-level enrichment of $m$ gene sets. It is important to note that $\mathbf{X}$ can hold either bulk tissue data (e.g., bulk RNA-sequencing) or single sample data (e.g., single cell RNA-sequencing) so the term sample can represent either a bulk tissue sample or individual cell depending on the context. This matrix of sample-level gene set scores can then be used in a wide range of subsequent computational tasks including unsupervised analyses like data visualization and supervised analyses like testing the association of each column of $\mathbf{S}$ with a given outcome variable, which generates results similar to those created by a population-level and supervised technique.

A number of uniset, unsupervised single sample gene set testing methods are currently available, which can be generally grouped into self-contained and competitive categories. Competitive single sample techniques like GSVA [12] and ssGSEA [13]) generate sample-level scores using a Kolmogorov-Smirnov (KS) like random walk statistic computed on the gene ranks within each sample, often following some form of gene standardization across the samples. AUCell [14], which is focused single cell transcriptomic data, also generates gene set scores based on gene ranks within each sample/cell using a simple "area under the curve" (AUC) metric to quickly quantify the enrichment of highly expressed genes within each evaluated set. In contrast to GSVA and ssGSEA, AUCell does not take into account gene set size or the distribution of gene expression values across all cells in the data set. Self-contained methods like PLAGE [15], PAGODA [16], the z-scoring method of Lee et al. [17], scSVA [18], Vision [19], and our VAM method [9] generate scores using only the data for genes in the set. Our development of the VAM technique was motivated by the poor performance of other single sample techniques on single cell transcriptomic data. Specifically, we found that existing techniques have poor classification performance (i.e., the ability to assign high gene set scores to cells with inflated expression of set genes) in the presence of sparsity and technical noise, and a high computational cost. The VAM method is a novel modification of the standard Mahalanobis multivariate distance measure [20] that generates cell-specific gene set scores which account for the inflated noise and sparsity of single cell RNA-sequencing (scRNA-seq) data. Because the distribution of the VAM-generated scores has an accurate gamma approximation under the null of uncorrelated technical noise, these scores can also be used for inference regarding pathway activity.

## Single sample gene set testing challenges

While the VAM technique offers a significant improvement in terms of computational and classification performance over other single sample methods, it has four important limitations:

1. **Sensitivity to differential correlation**: Most existing single sample gene set testing methods are designed to detect differences in mean value (i.e., set genes have higher expression than non-set genes in a competitive scenario) and struggle to identify biologically relevant patterns of differential correlation (i.e., the inter-gene correlation among set genes is higher than the correlation among non-set genes in a competitive scenario).

2. **Support for competitive $H_0$**: The VAM method, and other computationally efficient techniques like PLAGE [15] and the z-scoring method of Lee et al. [17], are self-contained methods that generate scores for a given gene set without considering the values of genes not in the set. These self-contained methods cannot directly detect competitive scenarios where the measured values of set genes differ from non-set genes in the same sample.

3. **Comparison of scores for different sets**: The scores generated by existing single sample methods can only be accurately compared across samples for a single set and not between sets. This limitation complicates many types of multivariate downstream analyses that attempt to jointly evaluate the scores for multiple sets. For example, it becomes challenging to determine which of several gene sets are more active/enriched in a given sample since the scores for different sets are not necessarily on the same scale.

4. **Computational cost**: The computational performance of VAM, while better than most existing methods, can still be significant on very large datasets. For VAM, and other computationally efficient self-contained methods, computational cost scales with the number of samples. As the price of single cell experimental methods continues to fall, the number of

cells in a typical dataset has grown substantially with tens-to-hundreds-of-thousands of cells now common. Projects that generate single cell data on samples from hundreds of separate patients will result in even larger total sample sizes. These very large single cell datasets motivate performance improvements beyond what can be obtained using techniques like VAM.

## Gene set testing based on reconstruction error

To address these challenges, we developed a new, and analytically novel, single sample gene set testing method called Reconstruction Set Test (RESET). RESET quantifies gene set importance based on the ability of genes in the set to reconstruct values for all measured genes. This reconstruction approach is effective at identifying scenarios where the mean expression and correlation of set genes is elevated in a group of cells relative to other measured genes. In particular, gene sets with elevated expression and/or correlation capture more of the overall signal in the dataset so are more effective in generating a reconstruction of the entire expression matrix than gene sets whose members have a similar distribution as other measured genes. RESET is realized using a computationally efficient randomized reduced rank reconstruction algorithm and can effectively detect patterns of differential abundance and differential correlation for both self-contained and competitive scenarios. The use of reconstruction error by RESET is distinct from standard approaches to gene set testing and has the potential to capture biological patterns not detectable using methods based on differences in mean expression. Unique among single sample methods, RESET generates both overall and sample-level scores for evaluated gene sets. Mathematical details of the RESET method and the evaluation design are outlined in the Methods section with some technical content in S1 Text. The Results section contains the simulation study and real data analysis results, which demonstrate that RESET provides superior classification accuracy at a lower computation cost relative to VAM and other popular single sample gene set testing approaches. An R implementation, which supports integration with the Seurat framework [21], is available in the RESET package on CRAN.

## Materials and methods

### Reconstruction Set Test (RESET)

The RESET method computes sample-specific and overall gene set scores from gene expression data using the error from a randomized reduced rank reconstruction. At a high-level, RESET takes as input two matrices:

- **X**: $n \times p$ matrix that holds the abundance measurements for $p$ genes in $n$ samples (or $n$ cells for single cell data). As outlined in the Methods section of S1 Text, RESET provides direct support for scRNA-seq data processed using the Seurat [22] framework using either log-normalization (i.e., log of 1 plus the unnormalized count divided by an appropriate scale factor for the cell) or the SCTransform method [23]. It should be noted that the RESET method itself (as defined by Algorithm 3 below) does not directly address the issues of data quality control (QC) or normalization/batch correction; it is assumed that appropriate QC and normalization is performed prior to execution of the RESET method.

- **A**: $m \times p$ matrix that represents the annotation of the $p$ genes in **X** to $m$ gene sets as defined by a collection from a repository such as the Molecular Signatures Database (MSigDB) [24] ($a_{i,j} = 1$ if gene $j$ belongs to gene set $i$). Note that some columns of **A** can sum to 0, i.e., certain genes in **X** may not belong to any of the evaluated gene sets.

RESET generates as output:

- **S**: $n \times m$ matrix that holds sample-specific gene set scores for each of the $n$ samples in **X** and $m$ gene sets defined in **A**.

- **v**: length $m$ vector that holds the overall scores for each of the $m$ gene sets defined in **A**.

The version of the RESET method implemented in the RESET R package and used to generate the results contained in this paper is defined in Algorithm 3 and visualized in Fig 1. This fully optimized version of RESET incorporates randomized numerical linear algebra (RNLA) [25] techniques and accepts a number of additional parameters (center, scale, num.pcs, pca. buff, pca.q, random.threshold, k, k.buff, q, test.dist, norm.type, per.var) whose function, motivation and interdependencies are fairly complex. To make this full method easier to understand, we start by defining a simplistic, and computationally inefficient, version of RESET, refine the simple version to use a more efficient reduced rank reconstruction, and then finally introduce the randomized RESET algorithm. The simplistic version of the RESET method, as detailed in Algorithm 1, uses all of the genes in each set to reconstruct the full matrix **X**, generates overall scores using the Frobenius norm (i.e., the square root of the sum of the squared matrix elements; see Martinsson and Tropp [25] for an overview of matrix norms) of the reconstruction error matrix, and generates sample-level scores using the Eucledian norm of the reconstruction error for the associated row (use of the $L_1$ norm is also supported by the RESET R package).

**Algorithm 1** Simplistic RESET

**Inputs:**

- **X:** $n \times p$ matrix that holds the abundance measurements for $p$ genes in $n$ samples.

- **A:** $m \times p$ matrix that holds the annotation of the $p$ genes in **X** to $m$ gene sets, $a_{i,j} = 1$ if gene $j$ belongs to gene set $i$.

**Outputs:**

- **S:** $n \times m$ matrix that holds sample-specific gene set scores for the $n$ samples in **X** and $m$ gene sets defined in **A**.

- **v:** length $m$ vector that holds the overall scores for each of the $m$ gene sets defined in **A**.

**Notation:**

- Let **X**[] represent a subsetting of the matrix **X** with **X**[$i$, $j$] the element in the $i^{th}$ row and $j^{th}$ column, **X**[$i$, ] the $i^{th}$ row, **X**[, $j$] the $j^{th}$ column, and **X**[**r**, **c**] the submatrix containing rows with indices in **r** and columns with indices in **c**.

1: $\mathbf{v} \in \mathbb{R}^m, \mathbf{S} \in \mathbb{R}^{n \times m}$ ▷ Initialize outputs **v** and **S**
2: **for** $i \in \{1, \ldots, m\}$ **do**
3: **c** = which(**A**[$i$, ] = 1) ▷ Create a length $l_i = \sum$ **A**[$i$, ] vector that holds the indices of the genes in set $i$
4: $\mathbf{X}_s$ = **X**[, **c**] ▷ Create a $n \times l_i$ subset of **X** for set $i$
5: $\mathbf{Q}_s$ = **qr**($\mathbf{X}_s$) ▷ Create an orthonormal basis for the column space of $\mathbf{X}_s$ via a QR decomposition

```
6:    X_r = Q_s Q_s^T X          ▷ Create a rank l_i reconstruction of X via pro-
   jection onto Q_s
7:    E = X - X_r          ▷ Create a reconstruction error matrix
8:    v_i = log_2(||X||_F/||E||_F)          ▷ Set the overall score for set i
   to the log_2 ratio of the Frobenius norms of X and E
9:    for j ∈ {1, ..., n} do
10:       S[j, i] = log_2(||X[j, ]||_2/||E[j, ]||_2)          ▷ Set the score
   for set i and sample j to the log_2 ratio of the Euclidean norms of
   row j of X and row j of E
   return S, v
```

**Algorithm 2** Reduced rank RESET

```
Outputs and notation are the same as for Algorithm 1. Inputs also
include:
```

  - *b*: Rank used for dimensionality reduction of **X**

  - *k*: Rank used for dimensionality reduction of each **X**_s

```
1: v ∈ ℝ^m, S ∈ ℝ^{n×m}          ▷ Initialize outputs v and S
2: X_c = scale(X)          ▷ Mean center, and optionally scale, columns
   of X
3: X_c = UΣV^T          ▷ Compute SVD of X_c
4: P = X_c V[, 1: b]          ▷ Project X_c onto top b PCs
5: for i ∈ {1, ..., m} do
6:    c = which(A[i, ] = 1)          ▷ Create a length l_i = ∑ A[i, ] vector
   that holds the indices of the genes in set i
7:    X_s = X[, c]          ▷ Create a n × l_i subset of X for set i
8:    Q_s = qr(X_s)[, 1: k]          ▷ Create a rank k orthonormal basis for
   the column space of X_s via a column-pivoted QR decomposition
9:    P_r = Q_s Q_s^T P          ▷ Create a rank k reconstruction of P via projec-
   tion onto Q_s
10:   E = P - P_r          ▷ Create a reconstruction error matrix
11:   v_i = log_2(||P||_F/||E||_F)          ▷ Set the overall score for set i
   to the log_2 ratio of the Frobenius norms of P and E
12:   for j ∈ {1, ..., n} do
13:       S[j, i] = log_2(||P[j, ]||_2/||E[j, ]||_2)          ▷ Set the score
   for set i and sample j to the log_2 ratio of the Euclidean norms of
   row j of P and row j of E
   return S, v
```

Although the simplistic version of RESET defined in Algorithm 1 captures the general structure of the method, it has several critical limitations:

- Computational cost can be significant if either **X** or the gene sets defined in **A** are large.

- Reconstruction of the full **X** matrix using all genes in a given set can produce scores in **v** and **S** that are dominated by noise when the biological signal in **X** has an effective rank that is much lower than the observed rank of **X**, which is common for genomic data.

- If gene sets defined by **A** have distinct sizes, the generated scores under a null scenario of completely random data in **X** will not be equivalent. In particular, scores will be elevated for larger sets as compared to smaller sets.

The limitations of the simplistic version of RESET can be effectively addressed by reconstructing a dimensionally reduced version of **X** using a dimensionally reduced version of **X**_s (the subset of **X** corresponding to each gene set). This approach can be efficiently realized by

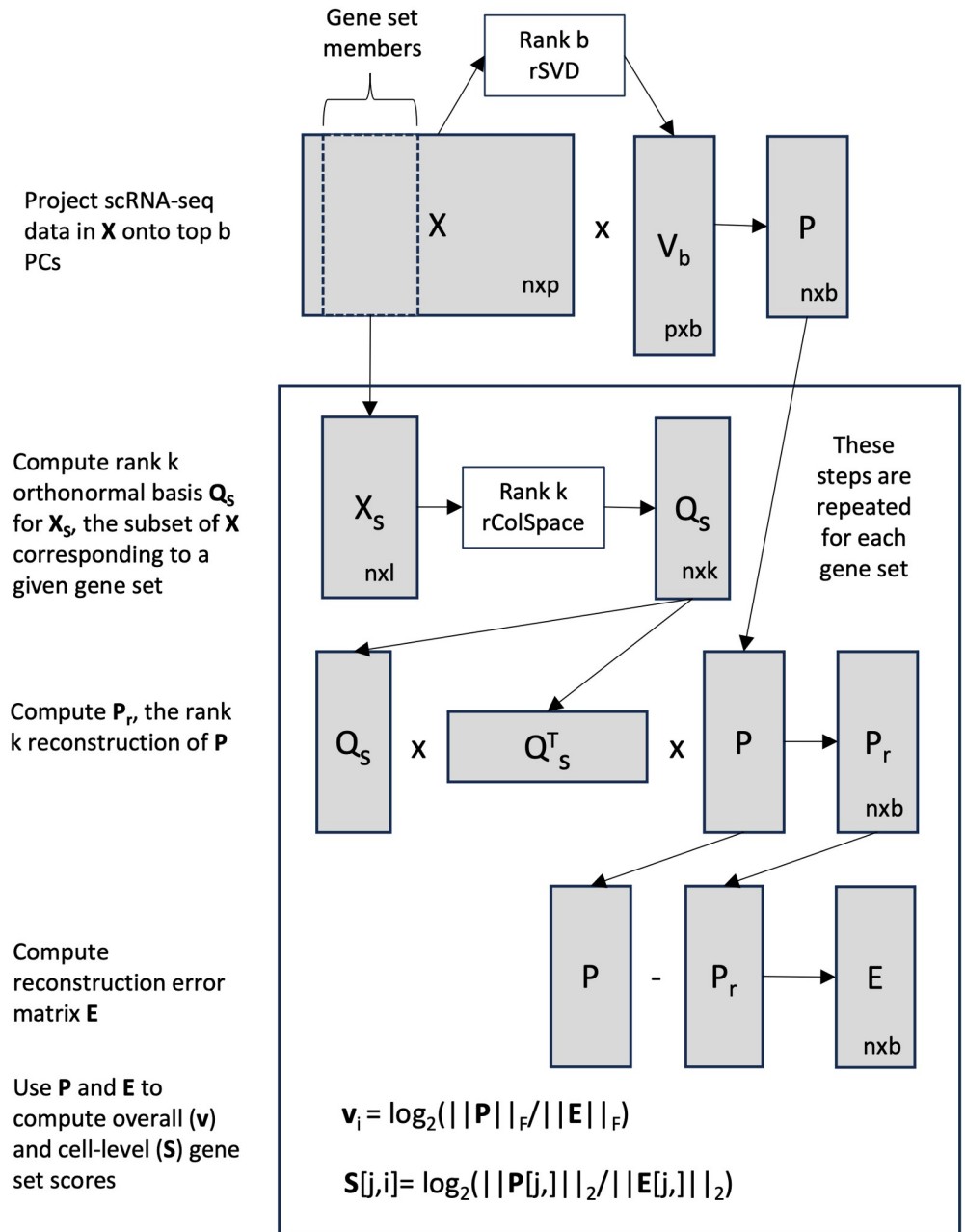

**Fig 1. Workflow representation of the randomized version of RESET as defined by Algorithm 3.**

projecting **X** onto the top $b$ principal components (PCs) where $b$ is close to the rank of the biological component of the data and then assessing how well this PC projection can be reconstructed using a rank $k$ basis for the column space of each **X**$_s$. In this case, PCA is used to reduce the dimensionality of **X** given the optimal properties of PCA (i.e., it generates the rank $b$ reconstruction with the minimal error as measurd by the Frobenius norm) and because PCA is often performed as a standard part of scRNA-seq processing pipelines. The optional scaling in step 2 has the same motivation as scaling in the context of PCA, i.e., it prevents genes with large variance from dominating the solution. Algorithm 2 defines this reduced rank version of

RESET. The Usage considerations section includes more details on how to select $b$ and $k$ and the impact of these parameters on method performance.

While the reduced rank RESET variant successfully addresses the key deficiencies of the simplistic version, it has two important limitations: 1) computational cost can still be significant, and 2) scores for gene sets of different sizes under a non-null scenario may not match user expectations. In terms of computational cost, the PCA and QR steps can be very expensive, even if truncated algorithms are used that halt after computing the top PCs/columns (e.g., the truncated PCA algorithm implemented in the irlba R package [26] or a truncated column-pivoted QR decomposition). Fortunately, the computational performance of these matrix decompositions can be dramatically improved by leveraging randomized numerical linear algebra (RNLA) [25, 27] techniques with only minimal loss of accuracy. Such RNLA methods have been successfully leveraged for the analysis of large genomic data matrices, e.g., scRNA-seq data, with data imputation via reduced rank reconstruction a key use case [28, 29]. The randomized version of RESET defined in Algorithm 3 (and visualized in Fig 1) relies on two underlying RNLA functions: a randomized technique for computing an orthonormal basis for the column space of a matrix and, building on that method, a randomized SVD algorithm. Note that the reconstruction of $\mathbf{P}$ via $\mathbf{Q}_s\mathbf{Q}_s^T\mathbf{P}$ is equivalent to creating a rank $k$ reconstruction via randomized SVD. The two RNLA techniques, which are defined in Algorithms A and B in S1 Text, follow the general structure of the randomized rangefinder and randomized SVD algorithms in Martinsson et al. [25]. At a high-level, these RNLA methods use random linear combinations of the columns of the original matrix to create a set of independent composite columns that can be used to approximate a reduced rank basis for the columns space. When the target reduced rank is much smaller than the overall rank of the matrix, these RNLA techniques provide a substantial performance improvement at the cost of a very minor error relative to a non-randomized decomposition. For clarity, Algorithm 3 omits parameters that allow for power iterations or delayed mean centering of $\mathbf{X}$ (see discussion below and the RESET R package documentation for details). Readers unfamiliar with RNLA should review Algorithms A and B in S1 Text. For more information on the theoretical and computation properties of these methods or the broader foundations/applications of RNLA, readers are encouraged to read the excellent survey by Martinsson and Tropp [25]. The paper by Erichson et al [30] associated with the *rsvd* R package provides a shorter introduction to these methods with a specific focus on their programmatic implementation and performance benefits relative to truncated algorithms.

Although the version of RESET defined by Algorithm 2 will yield equivalent scores under a null scenario for gene sets of different sizes, the scores produced under a non-null scenario may fail to match user expectations. For example, if two sets both include the same group of informative genes but one of the sets also includes an additional group of noise genes, the overall RESET scores for both sets will be equivalent when the reconstruction rank is less than the number of informative genes. However, given the behavior of standard gene set testing methods, users will likely expect that the overall score for the smaller set that only includes informative genes should be larger than the score for the set that also includes noise genes. To help address this issue, Algorithm 3 includes an option to compute per-variable reconstruction scores, i.e., scores that are divided by the scaled gene set size. Scaling in this case is performed by dividing the size of each set by the mean size across the entire collection. This is done to keep the per-variable scores on a similar scale as the unadjusted scores (scores for sets of average size will be unchanged). It is important to note that while the per-variable scores may better match user expectations in a non-null scenario, they will no longer be equivalent under the null.

**Algorithm 3** Randomized RESET

Outputs and notation are the same as for Algorithm 2. Inputs also include:

- *d*: Additional dimensions to compute using randomized methods.

- *random.threshold*: If the size of a given gene set is equal to or below this value, then the column space basis is computed using a deterministic method instead of the randomized technique defined in Algorithm A in S1 Text. When the deterministic method is used, the *per.var* flag is false and a non-random SVD is used for the initial matrix decomposition, this algorithm becomes equivalent to Algorithm 2.

- *per.var*: If true, scores are divided by the scaled gene set size.

1: $\mathbf{v} \in \mathbb{R}^m, \mathbf{S} \in \mathbb{R}^{n \times m}$ ▷ Initialize outputs $\mathbf{v}$ and $\mathbf{S}$
2: $\mathbf{X}_c$ = scale($\mathbf{X}$) ▷ Mean center, and optionally scale, columns of $\mathbf{X}$
3: $(\mathbf{U}, \boldsymbol{\Sigma}, \mathbf{V})$ = randomSVD($\mathbf{X}_c, b, d$) ▷ Compute randomized rank $b$ SVD of $\mathbf{X}_c$ using randomSVD function defined in Algorithm B in S1 Text
4: $\mathbf{P} = \mathbf{X}_c \mathbf{V}[, 1:b]$ ▷ Project $\mathbf{X}_c$ onto top $b$ PCs of $\mathbf{X}_c$
5: **for** $i \in \{1, ..., m\}$ **do**
6: $\mathbf{c}$ = which($\mathbf{A}[i, ] = 1$) ▷ Create a length $l_i = \sum \mathbf{A}[i, ]$ vector that holds the indices of the genes in set $i$
7: $\mathbf{X}_s = \mathbf{X}[, \mathbf{c}]$ ▷ Create a $n \times l_i$ subset of $\mathbf{X}$ for set $i$
8: **if** $l_i > random.threshold$ **then**
9: $\mathbf{Q}_s$ = randomColumnSpace($\mathbf{X}_s, k, d$) ▷ Compute an approximate rank $k$ orthonormal basis for the column space of $\mathbf{X}_s$ using the randomColumnSpace function defined in Algorithm A in S1 Text
10: **else**
11: $\mathbf{Q}_s$ = qr($\mathbf{X}_s$)$[, 1:k]$ ▷ Create a rank $k$ orthonormal basis for the column space of $\mathbf{X}_s$ via a column-pivoted QR decomposition
12: $\mathbf{P}_r = \mathbf{Q}_s \mathbf{Q}_s^T \mathbf{P}$ ▷ Create a rank $k$ reconstruction of $\mathbf{P}$ via projection onto $\mathbf{Q}_s$
13: $\mathbf{E} = \mathbf{P} - \mathbf{P}_r$ ▷ Create a reconstruction error matrix
14: $v_i = \log_2(||\mathbf{P}||_F / ||\mathbf{E}||_F)$ ▷ Set the overall score for set $i$ to the $\log_2$ ratio of the Frobenius norms of $\mathbf{P}$ and $\mathbf{E}$
15: **for** $j \in \{1, ..., n\}$ **do**
16: $\mathbf{S}[j, i] = \log_2(||\mathbf{P}[j, ]||_2 / ||\mathbf{E}[j, ]||_2)$ ▷ Set the score for set $i$ and sample $j$ to the $\log_2$ ratio of the Euclidean norms of row $j$ of $\mathbf{P}$ and row $j$ of $\mathbf{E}$
17: **if** *per.var* **then**
18: $v_i = v_i / (l_i / \bar{l})$ ▷ Divide overall score by the gene set length $l_i$ scaled by the average of all set lengths $\bar{l}$
19: $\mathbf{S}[, i] = \mathbf{S}[, i] / (l_i / \bar{l})$ ▷ Divide sample scores by the scaled gene set length $l_i / \bar{l}$
 **return S, v**

## Usage considerations

The randomized RESET method defined in Algorithm 3 and implemented in the RESET R package supports a number of parameters that enable customization of the method for different analysis scenarios. Three important use cases are the application of RESET to large sparse data sets, evaluation of gene set collections that contain sets whose size is close to the target

rank $k$, and evaluation of collections with a wide range of gene set sizes. Considerations for these scenarios, the selection of appropriate values for $k$ and $b$, deciding how to leverage the generated $\mathbf{S}$ and $\mathbf{v}$ scores, and the joint use of both RESET and VAM, are discussed below.

**Sparse X:** When $\mathbf{X}$ is large and sparse, which is typical of single cell data, it is usually represented by an optimized sparse matrix format (e.g., the sparse matrix support in the R *Matrix* package). In this case, mean centering of the columns of $\mathbf{X}$ will force conversion into a dense matrix format, which can have a significant impact on both memory usage and computational complexity for subsequent matrix operations. To avoid this performance penalty, it is desirable to only mean center a subset of $\mathbf{X}$ containing the data needed for gene set testing. How this scenario can be handled for RESET depends on whether the method is being executed via the Seurat framework interface (i.e., the *resetForSeurat()* function) or directly via either the *resetViaPCA()* or *reset()* functions. In both cases, RESET can be executed such that mean centering is only applied to a subset of $\mathbf{X}$ containing the genes that belong to the evaluated gene sets, i.e., it does not force the entire $\mathbf{X}$ matrix into a dense format. This mean centering is performed one gene set at a time within the inner loop of Algorithm 3. When RESET is executed via the Seurat framework interface, it is assumed that PCA has already been performed on a scaled and mean centered version of the normalized scRNA-seq data. Because Seurat by default only applies mean centering to a subset of the scRNA-seq data corresponding to the genes with the largest biological variance, the memory and performance impact is less severe. The Seurat wrapper passes in the unscaled normalized scRNA-seq matrix to the *reset()* function with parameters set so that mean centering of $\mathbf{X}$ is only performed on the columns for each gene set. If RESET is executed via the *resetViaPCA()* function, this delayed mean centering can be enabled by setting the *center* parameter to false, which will result in $\mathbf{X}$ being projected onto the uncentered PCs and centering performed on just the PC projections and subsets of $\mathbf{X}$ corresponding to each gene set. If RESET is executed via the *reset()* function, then users have full control over mean centering behavior (see the R package documentation and vignettes for more details).

**Gene set size is close to target $k$:** The computational benefit of the randomized column space basis generator detailed in Algorithm 3 is only meaningful if the number of columns in the input matrix is at least 3 times larger than the target rank $k$ [30]. This means that use of randomization for gene sets whose size is less than $\sim 3k$ will incur an accuracy penalty without any improvement in execution time. In this case, it is desirable to instead use the column-pivoted QR decomposition approach. The randomized RESET method detailed in Algorithm 3 supports this flexibility via the *random.threshold* argument. Randomization can be required for all evaluated sets by setting *random.threshold* to a value that is less than the minimal gene set size. Similarly, use of the deterministic column-pivoted QR decomposition for all gene sets can be achieved by setting *random.threshold* to the maximum gene set size.

**Large variability in gene set sizes:** Although the standard RESET algorithm will generate scores under the null that are similar for gene sets of different sizes, this equivalence may not hold under the alternative. In that case, users may want to set the *per.var* parameter to true to scale the generated scores by set size. It is important to note that the per-variable adjustment will result in non-equivalent scores under the null. It is also important to note that this issue is typically relevant only for the overall scores; most uses of the sample-level scores, e.g., differential expression testing, are insensitive to whether or not the per-variable adjustment is performed.

**Selecting appropriate values for $b$ and $k$:** How to select the target rank for reduced rank matrix decompositions is a long standing problem in applied mathematics. Although the RESET method does not directly address the rank selection problem and leaves specification of these parameters to users, there are a number of established approaches that can be followed

when determining appropriate values for *b* and *k*. Most of these techniques select the target rank based on the distribution of singular values using either a heuristic criteria, e.g., the elbow method, a model-based threshold, e.g., use of the random matrix theory-based eigenvalue null distribution [31], or a resampling technique, e.g., the JackStraw procedure used in the Seurat framework [32]. The simulation and real data results presented in the Results section use approximate, and likely non-optimal, values for *b* and *k*; performance of RESET in these cases could probably be improved through use of a more sophisticated rank selection method. It is also useful to keep in mind the impact that large or small values of parameters may have on method performance:

- *b*: Large values of *b* risk overestimating the rank of the biological component of **X** and thereby including the reconstruction of noise in the generated scores. In contrast, small values of *b* risk underestimating the biological rank of *X* with the consequence that reconstruction is measured on just a portion of the biological signal present in the data. In both cases, the RESET scores will be a less accurate reflection of how well each set captures the biological signal in the data with large *b* adding noise to the scores and small *b* adding bias.

- *k*: Large values of *k* risk including noise components of the evaluated gene sets when performing the reconstruction calculation. Because RESET uses a fixed value of *k* for all sets, this may pose a particular problem when comparing the scores of small and large gene sets, i.e., the reconstruction score is more likely to incorporate noise for the small sets than for the large sets. Small values of *k* risk ignoring some of the biological component of the evaluated gene sets. Similar to the impact of large *k* values, this will specifically impact the comparison of scores for different sized sets with large sets more likely to suffer score deflation because only a portion of the biological signal was considered. Supporting the use of set-specific *k* values (e.g., base *k* on gene set size), would help mitigate these issues and is something we may explore in future versions of RESET.

**Using the S and v scores:** Unique among single sample gene set testing methods, RESET generates both sample-level scores in the **S** matrix and overall scores in the **v** vector. The sample-level scores provide the most flexibility and can be used in place of gene abundance data in a wide range of subsequent statistical analysis, e.g., differential expression analyses, regression modeling, clustering, visualization, etc. Although the overall gene set scores offer less general utility, they still provide distinct information regarding the overall biological signal in the data and can be leveraged for filtering or weighting of pathway-based models.

**Joint analysis using RESET and VAM:** Although RESET provides superior performance relative to other single sample methods for data structures involving differential correlation, techniques like VAM are optimal for detecting patterns of differential abundance in the absence of differential correlation. The PBMC results shown below are a good example of the distinct output that can be generated by RESET and VAM on real scRNA-seq data. Given these factors, the use of both RESET and a differential abundance test like VAM may be motivated for many analysis scenarios.

## Comparison methods

The relative performance of the RESET method was assessed on simulated and real scRNA-seq data (see the Methods section in S1 Text for details on the evaluation framework). For the simulation study, three different versions of RESET (RESET.det, RESET.ran and RESET.per-var) were evaluated based on the setting of the *random.threshold* and *per.var* parameters. For RESET.det, which stands for deterministic RESET, *random.threshold* was set to 90, which

forces the use of the column-pivoted QR decomposition, and *per.var* was set to false. RESET. pervar used the same *random.threshold* value but set *per.var* to true to adjust the scores according to gene set size. For RESET.ran, which stands for randomized RESET, *random.threshold* was set to 9, which forces the use of the randomized column space basis generator (Algorithm A in S1 Text), and *per.var* was set to false. All three RESET variants were compared against the output from our previous developed VAM method [9] and three other single sample techniques: GSVA [12], ssGSEA [13], and PLAGE [15]. GSVA and ssGSEA were selected because of their widespread use in the field and because they incorporate both competitive and self-contained features (as opposed to VAM, which is purely self-contained). PLAGE was selected as example of a self-contained PCA-based technique that is sensitive to patterns of differential correlation. PLAGE generates gene set scores using the projection of each sample onto the first PC of the sub-matrix corresponding to the evaluated gene set (PAGODA [16] generates cell-level gene set scores uses a similar approach but the overall process is more complex, requires a specific normalization and first assesses overall PC significance using a Tracey-Widom test on the associated eigenvalue). For VAM, we used the implementation in version 1.0.0 of the VAM R package from CRAN. For GSVA, ssGSEA and PLAGE, we used the implementations available in version 1.46.0 of the GSVA R package from Bioconductor. Unless otherwise noted, all the comparison methods were executed using default parameter values.

## Results and discussion

### Sample-level classification performance

To compare the performance of RESET against our previously developed VAM technique and the GSVA, ssGSEA and PLAGE methods, we measured the classification performance (i.e., the ability of each method to generate high scores for cells that have higher mean expression and/ or non-zero correlation between genes in a specific set) on simulated scRNA-seq data. Specifically, scRNA-seq data was simulated as independent negative binomial counts with a fixed dispersion and random means generated from a shifted exponential distribution. The genes included in the first evaluated gene set are considered informative with counts generated using a higher expected negative binomial mean. Finally, the counts corresponding to a subset of the cells for the informative genes are given a flexible correlation structure and inflated mean values to produce a block of counts with both differential correlation and differential expression relative to the rest of the simulated matrix. Six different simulation parameters were varied to generate a wide range of data structures:

- **Mean inflation**: The factor by which the mean of the negative binomial distribution used to simulate scRNA-seq counts is inflated for the block of informative cells and genes.

- **Correlation**: The pairwise correlation between the simulated counts for the informative genes in the informative cells.

- **Set size**: The number of genes in the evaluted gene set(s). The genes in the first set are considered informative and are simulated with correlation and elevated mean values in the informative cells.

- **Number of informative cells**: The number of cells for which counts of the informative genes are simulated with correlation and inflated means.

- **Mean**: The baseline mean of the negative binomial distribution used to simulate counts for the informative genes in all cells.

- **Rate**: The rate of exponential distribution that is used to generate random variation in the mean of the negative binomial distribution used to simulate counts for the informative genes in all cells.

For full details on these parameters and the associated simulation framework, please see Section 1.3 in S1 Text, which includes a link to the associated R logic. Fig 2 illustrates the relative classification performance (as measured by the area under the receiver operating characteristic curve (AUC)) of RESET.det, RESET.ran, VAM, GSVA, ssGSEA, and PLAGE for the block design and a single gene set across a range of mean inflation, inter-gene correlation, gene set size and number of informative cells (i.e., number of cells for which gene set values are inflated and/or correlated). Fig 3 shows the results for a block model with a more complex correlation structure, and Figs 4 and 5 provide results for the pure self-contained and pure competitive scenarios using a single gene set or a collection of five disjoint and equally sized gene sets, respectively. All of these figures display the average AUC (and standard error of the mean via error bars) for 50 simulated data sets generated for each distinct combination of parameter values. Importantly, both the deterministic and randomized versions of the RESET method provide superior classification performance relative to VAM, GSVA, ssGSEA, and PLAGE across nearly the full range of evaluated parameter values for all three simulation designs with significant relative performance benefits for the pure self-contained and pure competitive cases. Two general trends are important to note:

- **Performance relative to mean inflation of informative genes in informative cells**: As expected, classification performance of nearly all of the evaluted methods improves as the mean of informative genes is increased when there is just a single block of informative genes and informative cells. This is visualized in the first panel of Figs 2 and 3. By contrast, classification performance is insensitive to mean inflation when either all cells are informative for a single gene set (the pure competitive model whose results are shown in Fig 5) or all genes are informative for a subset of cells (the pure self-contained model whose results are shown in Fig 4). For the pure self-contained model, this is due to the fact that the log-normalization process will largely eliminate the differential expression signature. For the pure competitive model, this is due to the fact that self-contained methods will see no differential expression signature (i.e., the mean of informative genes is inflated in all cells) and, for competitive methods, the fact that the employed normalization processes eliminate mean differences between genes.

- **Performance relative to the level of correlation between informative genes in informative cells**: In contrast to the other evaluated methods, RESET performance improves noticably as the correlation between informative genes increases. This can be seen in the second panel of Figs 2–5. For RESET, this performance trend is due to the fact that as a subset of variables becomes more correlated relative to all variables in a given matrix, those variables will become increasingly associated with a reduced rank representation of the matrix. In the extreme case where there is just a single informative principal component for a matrix (i.e., the non-random signal in the matrix is due to a single correlated block in the population covariance matrix), the set of variables in the correlated block can effectively reconstruct the entire matrix. The reason the other evaluated methods do not see a similar trend is due to both the self-contained nature VAM and PLAGE and, for all evaluated techniques except for PLAGE, a focus on differential abundance vs. differential correlation.

These two trends are discussed in more detail below in the context of Figs 2–5.

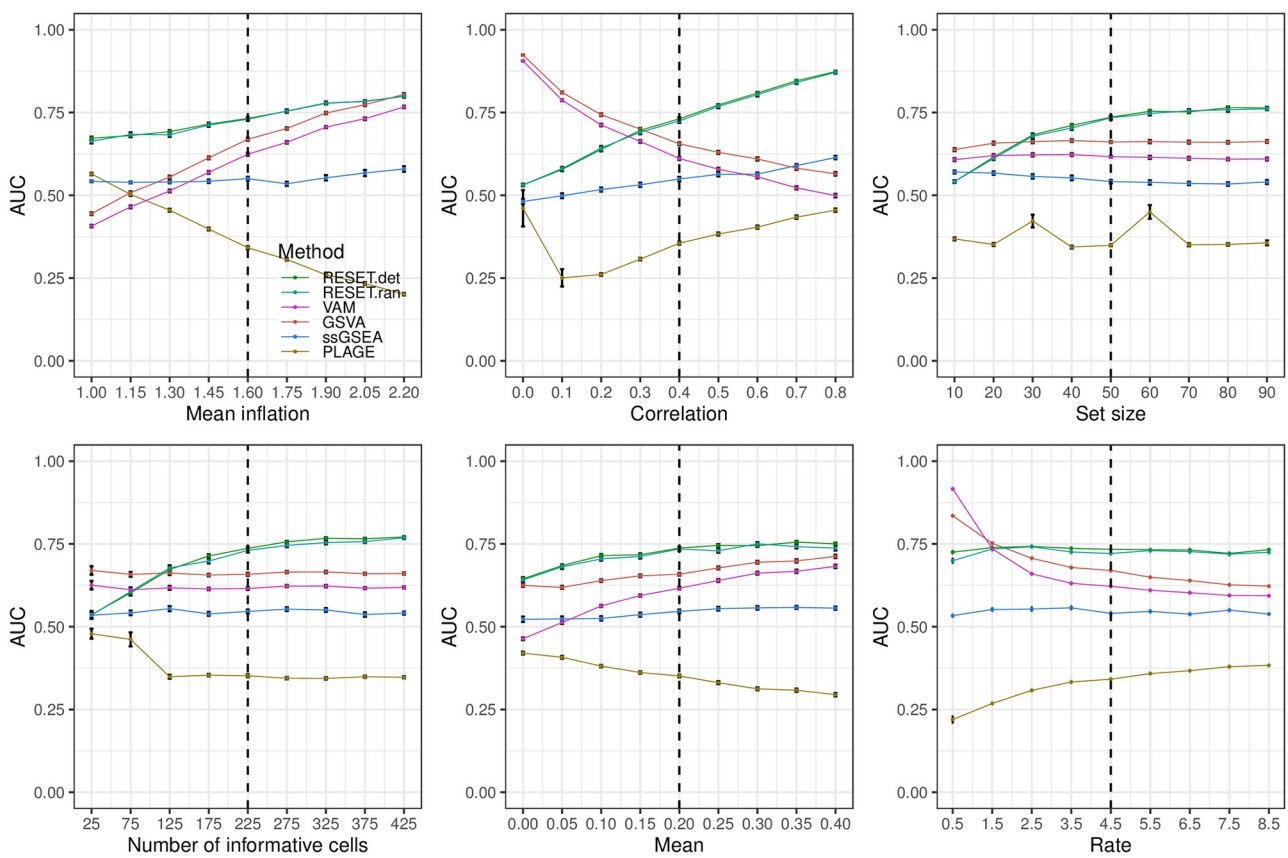

**Fig 2. Classification performance of RESET.det, RESET.ran, VAM, GSVA, ssGSEA, and PLAGE on scRNA-seq data simulated according to the block design for a single gene set as detailed in the Methods section in S1 Text.** Each panel illustrates the relationship between the area under the receiver operating characteristic curve (AUC) and one of the simulation parameters. The vertical dotted lines mark the default parameter value used in the other panels. Error bars represent the standard error of the mean.

The results for the block simulation design using a single gene set are visualized in Fig 2 and indicate that the two versions of RESET have very similar performance with the deterministic variant generating slightly better AUC values than the randomized variant, as expected. RESET provides measurably better classification performance than the other methods across all parameter settings except for the low correlation case, low number of informative cells case, and low rate case (i.e., the rate for the shifted exponential distribution of the random negative binomial mean for each gene), for which VAM and GSVA have the best performance with GSVA slightly higher than VAM. Notably, ssGSEA generates nearly null AUC values of just slightly above 0.5 for this simulation design. Also noticeable is the fact that AUC values for PLAGE are either close to null or below 0.5, which is due to the fact that the sign of PC loadings is arbitrary. As expected, performance of RESET improves with higher mean inflation, higher inter-gene correlation, larger gene set size, and increased number of informative cells. Consistent with limitation 1 (*sensitivity to differential correlation*), RESET provides a particularly large performance benefit in the high correlation scenario. Interestingly, VAM and GSVA performance decreases as inter-gene correlation is increased, which follows from both the impact of correlation on sparse count values (i.e., this will generate many cells with 0 values for most genes in the set) and the use by VAM of a correlation-breaking permutation to determine a null score distribution. Block design results using a collection of five overlapping and

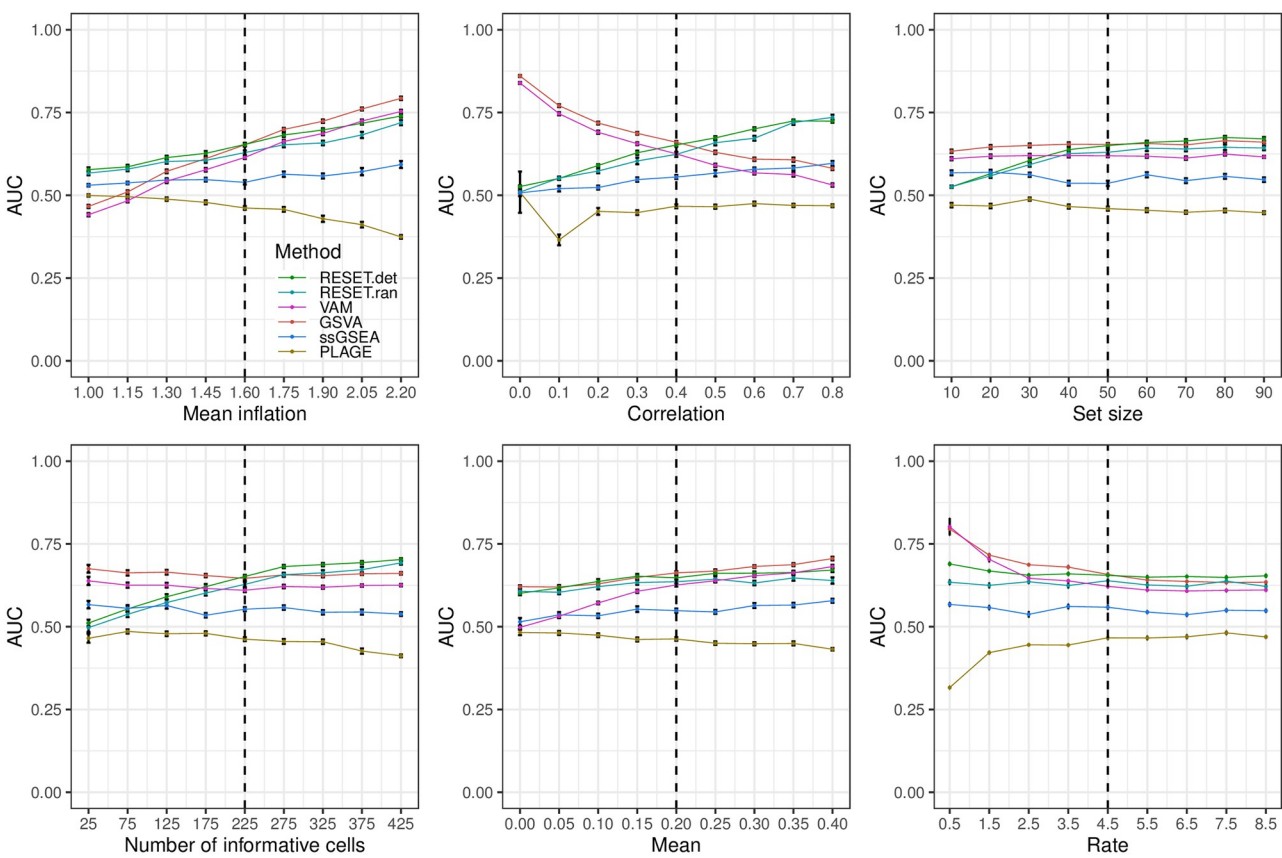

**Fig 3. Classification performance of RESET.det, RESET.ran, VAM, GSVA, ssGSEA, and PLAGE on scRNA-seq data simulated according to the complex design for a single gene set as detailed in the Methods section of** S1 Text**.** Each panel illustrates the relationship between the area under the receiver operating characteristic curve (AUC) and one of the simulation parameters. The vertical dotted lines mark the default parameter value used in the other panels. Error bars represent the standard error of the mean.

unequally sized gene sets (shown in Fig A in S1 Text) are generally consistent with the single set case with two exceptions: 1) ssGSEA has AUC values significantly below 0.5 (see comments on the competitive scenario illustrated in Fig 5 below), and 2) VAM AUC values are slightly higher than GSVA AUC values.

The results for the complex design, i.e., a model where the first gene set is split into two groups of genes that have distinct patterns of mean inflation and correlation, are shown in Fig 3. This model captures the case where the expression data for a gene set has two non-random PCs rather than just a single identifiable PC. The statistical model used to simulate data for this design is detailed Section 1.3 in S1 Text. For this simulation model, the RESET method performance is very similar to VAM and GSVA across most of the tested parameter values with a clear benefit only present at high correlation and low mean inflation values. This follows from the fact that only a subset of the gene set members are associated with the informative cells. One important consequence of this model is that the signal that separates informative from non-informative cells is represented by PC 2 of the gene set sub-matrix, i.e., the PC associated with the second largest eigenvalue. Since RESET was set to use a rank 10 representation of the gene set for this analysis, it can capture this signal, which is not the case for techniques like PLAGE and PAGODA only use the first PC to generate scores. This is reflected in the near null performance of PLAGE for this scenario.

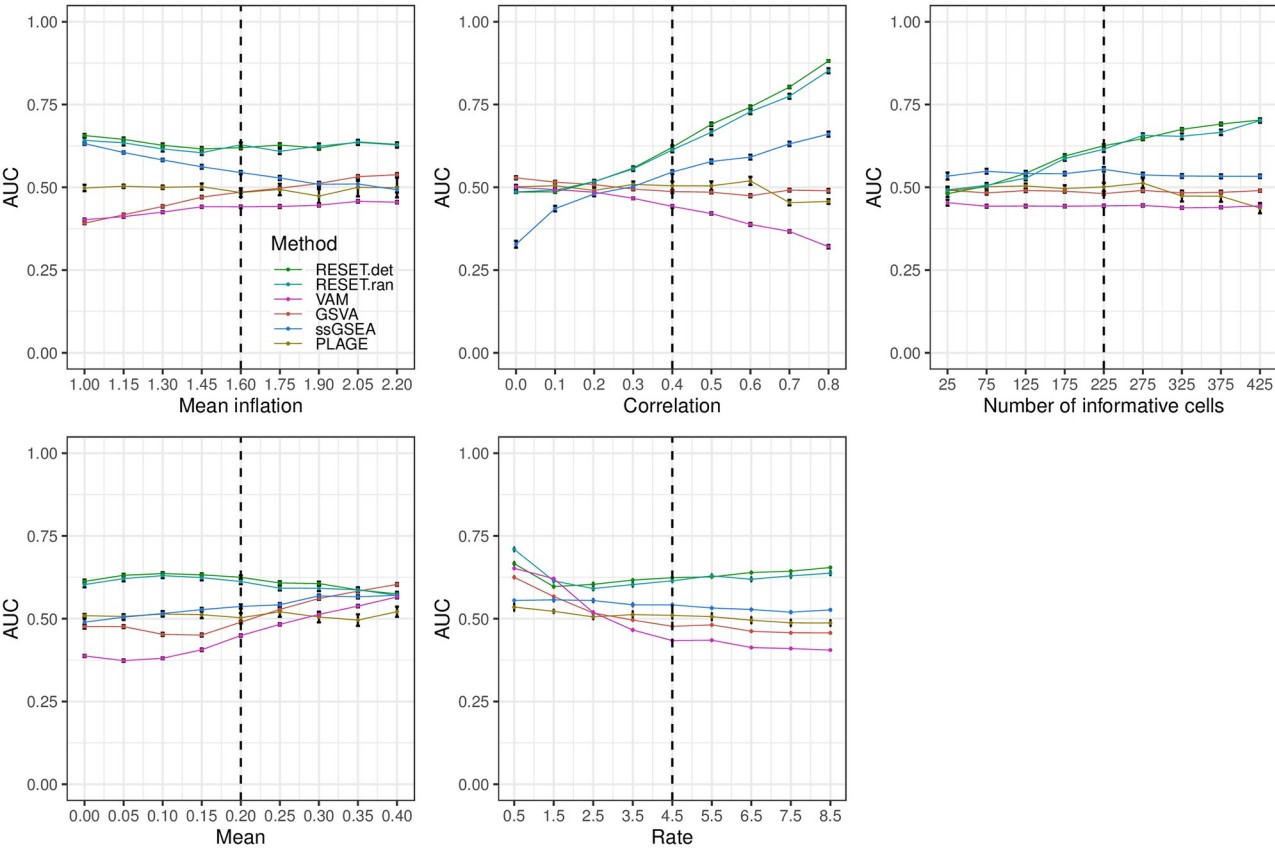

**Fig 4. Classification performance of RESET.det, RESET.ran, VAM, GSVA, ssGSEA, and PLAGE on scRNA-seq data simulated according to the pure self-contained design for a single gene set as detailed in the Methods section in S1 Text.** Each panel illustrates the relationship between the area under the receiver operating characteristic curve (AUC) and one of the simulation parameters. The vertical dotted lines mark the default parameter value used in the other panels. Error bars represent the standard error of the mean.

The results for the pure self-contained design, i.e., a model where all genes are included in the enriched set, are shown in Fig 4. It should be noted that the log-normalization process will largely eliminate the differential expression signature in this case, which explains the overall poor performance of VAM, GSVA, ssGSEA and PLAGE and the fact that method performance is relatively insensitive to mean inflation. For this simulation model, the RESET method is dominant across a wider range of parameter values than for the block design. GSVA has nearly null performance for this model, which is expected given the competitive aspect of this technique, i.e., it compares values for genes in the set to genes not in the set for a single cell. Similar to the block design, the performance benefit of RESET is especially large for the high correlation case, which provides additional confirmation of limitation 1 (*sensitivity to differential correlation*). Interestingly, the performance of ssGSEA, while still close to null for most parameter values, is better than for the block design with noticably improved performance for at higher inter-gene correlation values. Although this type of data structure does not reflect a realistic biological scenario, it helps highlight the self-contained vs. competitive attributes of the various methods.

The results for the pure competitive design, i.e., a model where genes in the set have inflated/correlated counts for all cells, are shown in Fig 5. While the block and self-contained models could be assessed using only the scores for a single gene set, the pure competitive case

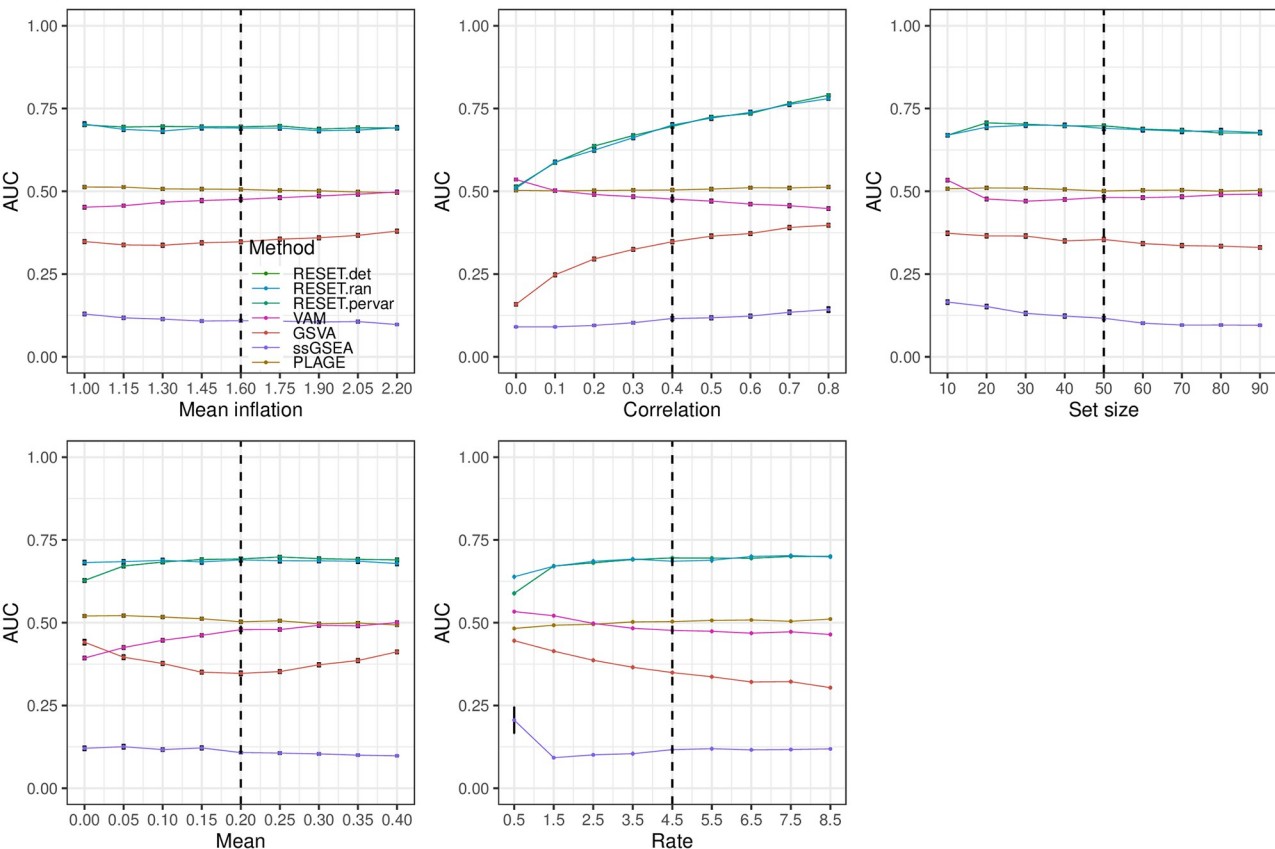

**Fig 5. Classification performance of RESET.det, RESET.ran, RESET.pervar, VAM, GSVA, ssGSEA, and PLAGE on scRNA-seq data simulated according to the pure competitive design for disjoint and equally sized genes as detailed in the Methods section in S1 Text.** Note that RESET.det and RESET.pervar have identical performance in this case since the gene sets all have the same size. Each panel illustrates the relationship between the area under the receiver operating characteristic curve (AUC) and one of the simulation parameters. The vertical dotted lines mark the default parameter value used in the other panels. Error bars represent the standard error of the mean.

requires comparison between the scores for the enriched set and the scores for non-enriched sets (four disjoint and equally sized non-enriched sets were used for this simulation). The results for this model are quite dramatic, with RESET providing good classification performance across all parameter values, VAM, GSVA and PLAGE yielding close to null values and ssGSEA generating AUC values significantly below 0.5. For VAM and PLAGE, the null performance is expected given the self-contained nature of these tests, i.e., since all cells are enriched for the gene set, there is no self-contained signature to detect; these results are consistent with limitation 2 (*support for competitive $H_0$*). For GSVA, the approximately null results are consistent with the fact that the method converts the values for each gene into quantiles using an empirical density estimate for the gene, which erases the differential expression signature in this case. For ssGSEA, the very low AUC values are surprising. If a classifier consistently generated low AUC values, it would be feasible to turn it into a high AUC classifier simply by inverting the score ordering. In this case, however, ssGSEA is generating both high and low AUC values for different simulation scenarios so attempting to improve performance for the pure competitive design would break performance for other designs. As outlined in limitation 3 (*comparison of scores for different sets*), the scores generated by GSVA and ssGSEA on different sets are not intended to be directly comparable, i.e., they can be on very different scales for sets

with largely the same pattern of expression. While standardization of the scores for each set can address this feature, such standardization would prevent detection of a pure competitive pattern. Competitive design results using a collection of five overlapping and unequally sized gene sets (shown in Fig C in S1 Text) have a similar pattern as the disjoint collection case with two main differences: 1) the relative performance benefit of RESET is smaller, and 2) the AUC values for ssGSEA are not a significantly below 0.5.

## Overall classification performance

We also evaluated RESET according to overall classification performance, i.e., can the method generate high overall scores for sets that have differential expression/correlation in a subset of the samples? The results for this evaluation are visualized in Fig 6 for the block design using a collection of five disjoint and equally sized gene sets. Because only RESET provides both overall and sample-level scores, just the RESET.det, RESET.ran and RESET.pervar variants are shown. Although it is challenging to intepret these results given the lack of comparative methods, they demonstrate very accurate performance for this simulation design across almost all parameter values. Since all gene sets are the same size, both RESET.det and RESET.pervar have equivalent performance with both slightly better than RESET.ran. Overall results using a collection of five overlapping and unequally sized gene sets (shown in Fig B in S1 Text) are very

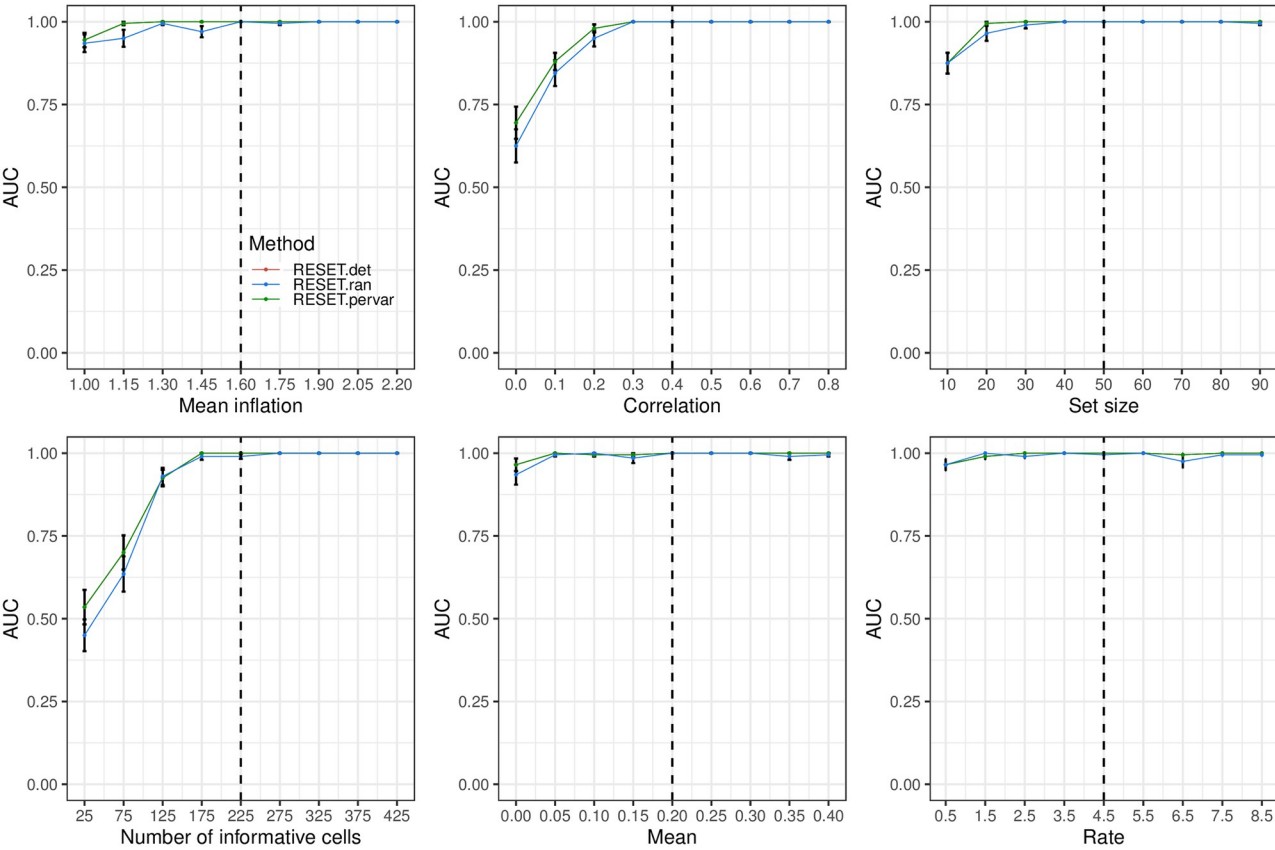

**Fig 6. Overall classification performance of RESET.det, RESET.ran and RESET.pervar on scRNA-seq data simulated according to the block design with disjoint and equal sized gene sets as detailed in the Methods section in S1 Text.** Each panel illustrates the relationship between the area under the receiver operating characteristic curve (AUC) and one of the simulation parameters. The vertical dotted lines mark the default parameter value used in the other panels. Error bars represent the standard error of the mean.

similar to the disjoint collection case. Although we had expected RESET.pervar to have better performance in this scenario, the AUC values for RESET.det and RESET.pervar variants show only minor differences.

## Computational efficiency

Table 1 displays the relative execution time of GSVA, ssGSEA, VAM and PLAGE as compared to the fully randomized version of RESET (i.e., "RESET.ran"). Relative times are shown for the analysis of the simulated data sets (2,000 cells and 500 genes) used to generate the classification results shown in Fig 2 (see Fig D in S1 Text for performance on this simulated data as a function of gene set size), for the analysis of the 3k cell PBMC scRNA-seq data set using the the Bio-Carta (C2.CP.BIOCARTA) collection from the Molecular Signatures Database (MSigDB) [24] (see the Human PBMC analysis section for detailed results), for the analysis of the 8.6k cell human cord blood scRNA-seq data set for the BioCarta collection (see Section 2.5 in S1 Text for detailed results), for the analysis of the 11.8k cell mouse brain scRNA-seq data set using the MSigDB Gene Ontology biological process (C5.BP) pathway collection (see the Mouse brain cell analysis section for detailed results), and for the analysis of the very large 242k cell Mouse Cell Atlas (MCA) [33] scRNA-seq data set using a single gene set containing the first 50 genes. Since the R implementation of GSVA and ssGSEA force the conversion of the gene expression matrix into a dense format, memory limitations prevented execution of these methods on the MCA data (PLAGE also forces conversion into a dense matrix but is self-contained so this is just performed for the 50 genes in the analyzed set). For more details on the PBMC, cord blood, mouse brain and MCA data sets and processing pipeline, see the Methods section in S1 Text. Although absolute execution times will vary significantly between different computing platforms, on a standard laptop RESET takes around 5.2 seconds to analyze the PBMC data for the MSigDB BioCarta collection, around 11 seconds to analyze the cord blood data for MSigDB BioCarta collection, and around 13 minutes to analyze the mouse brain data for MSigDB C5.BP collection.

Consistent with limitation 4 (*computational cost*), the RESET technique is two to four times as fast as VAM and nearly two to three orders-of-magnitude faster than GSVA and ssGSEA across both the simulated and real scRNA-seq data. Although RESET is consistently faster than PLAGE, the performance benefit is very modest for the real scRNA-seq datasets, which is consistent with the limited performance benefits of RNLA for the number of variables contained in the gene set sub-matrices as well as the fact that reconstruction of the entire scRNA-seq matrix and computation of reconstruction error is only performed by RESET. One interesting result is the substantially worse relative performance of GSVA and ssGSEA on the cord blood data as compared to the similarly sized mouse brain data. This result can be understood from the fact that execution time for GSVA and ssGSEA is primarily driven by the size of the

**Table 1. Ratio of execution time for the GSVA, ssGSEA, VAM, and PLAGE methods to the execution time for RESET on simulated scRNA-seq data, the PBMC scRNA-seq data set for the MSigDB C2.CP.BIOCARTA collection, the cord blood scRNA-seq data set for the MSigDB C2.CP.BIOCARTA collection, the mouse brain scRNA-seq data set for the MSigDB C5.BP collection, and the Mouse Cell Atlas for a single synthetic gene set.** For the real scRNA-seq data, RESET was executed using the parameters specified in the Methods section in S1 Text. For the simulated data, execution times are relative to the fully randomized version (i.e., "RESET.ran") as detailed in the Methods section in S1 Text.

| | Simulated | PBMC | Cord blood | Mouse brain | MCA |
|---|---|---|---|---|---|
| GSVA | 416.87 | 97.83 | 407.45 | 11.34 | - |
| ssGSEA | 81.28 | 71.17 | 194.59 | 83.82 | - |
| VAM | 4.36 | 2.45 | 4.18 | 11.16 | 26.57 |
| PLAGE | 11.47 | 1.64 | 1.84 | 1.23 | 1.36 |

input scRNA-seq matrix whereas for RESET it is primarily a function of number of gene sets. In this example, the count matrices have similar sizes but BioCarta contains many fewer gene sets than MSigDB C5.BP. As shown in Fig D in S1 Text, the randomized version of RESET only provides a noticable performance benefit relative to the deterministic version of RESET when the gene set size is roughly four to five times larger than the target rank $k$, which is consistent with findings by Erichson et al [30].

In addition to computational time, differences in memory requirements are an important practical consideration. For all of the methods, memory consumption is primarily a function of the target dataset with differences between the evaluated methods largely due to whether or not the techniques leverage a sparse matrix framework. VAM and RESET both maintain the sparse matrix representation used by frameworks like Seurat, so have very similar memory requirements as a standard Seurat pipeline. GSVA, ssGSEA, and PLAGE on the other hand, convert the sparse matrix into a dense format so can have dramatically larger memory requirements depending on the size and inherent sparsity of the analyzed data. It is also worth noting that certain operations like mean centering will eliminate sparsity so can have a very large memory cost irrespective of the gene set testing method (see the discussion for the "Sparse X" topic in the Usage considerations section).

## Human PBMC analysis

As detailed in the Methods section of S1 Text, we applied the RESET method and the comparison techniques to the 10x 2.7k human PBMC scRNA-seq data set. Fig E in S1 Text is a reduced dimensional visualization of the 2,638 cells remaining after quality control filtering. Cluster cell type labels match the assignments in the Seurat Guided Clustering Tutorial. For this analysis, we looked at both the overall and cell-specific pathway scores generated by RESET. Table 2 lists the top 20 BioCarta pathways according to the overall RESET score without the per-variable adjustment and Table A in S1 Text contains the top 20 with the per-variable adjustment. Although the per-variable adjustment has a noticible impact on pathway ranking (e.g., only 5 of the pathways listed in Table 1 are also present in Table A in S1 Text), both the unadjusted and per-variable RESET scores accurately reflect the immune cell source of this scRNA-seq data set with nearly all of the top 20 pathways according to either score type having an association with a specific immune cell type or immune signaling pathway, e.g., CSK (activation of Csk by cAMP-dependent protein kinase inhibits signaling through the T cell receptor), FCER1 (fc epsilon receptor I signaling in mast cells), MHC (antigen processing and presentation), and CTL (cytotoxic lymphocyte mediated immune response against target cells).

**Table 2. Top 20 BioCarta pathways according to overall RESET score for the PBMC data set.**

| Rank | Pathway | RESET score | Rank | Pathway | RESET score |
|---|---|---|---|---|---|
| 1 | BIOCARTA-CSK-PATHWAY | 0.138 | 11 | BIOCARTA-TCRA-PATHWAY | 0.061 |
| 2 | BIOCARTA-FCER1-PATHWAY | 0.108 | 12 | BIOCARTA-IL10-PATHWAY | 0.059 |
| 3 | BIOCARTA-MHC-PATHWAY | 0.107 | 13 | BIOCARTA-SPPA-PATHWAY | 0.057 |
| 4 | BIOCARTA-CTL-PATHWAY | 0.101 | 14 | BIOCARTA-BLYMPHOCYTE-PATHWAY | 0.050 |
| 5 | BIOCARTA-UCALPAIN-PATHWAY | 0.095 | 15 | BIOCARTA-INFLAM-PATHWAY | 0.045 |
| 6 | BIOCARTA-TCR-PATHWAY | 0.092 | 16 | BBIOCARTA-GPCR-PATHWAY | 0.045 |
| 7 | BIOCARTA-THELPER-PATHWAY | 0.086 | 17 | BIOCARTA-BCR-PATHWAY | 0.043 |
| 8 | BIOCARTA-TCYTOTOXIC-PATHWAY | 0.067 | 18 | BIOCARTA-EICOSANOID-PATHWAY | 0.042 |
| 9 | BIOCARTA-IL17-PATHWAY | 0.066 | 19 | BIOCARTA-CARM1-PATHWAY | 0.041 |
| 10 | BIOCARTA-CHEMICAL-PATHWAY | 0.062 | 20 | BIOCARTA-FMLP-PATHWAY | 0.040 |

To visualize how well the RESET scores for the BioCarta pathways capture the overall transcriptomic signal in the original scRNA-seq data, one can compute a UMAP projection on the RESET scores. Fig F in S1 Text displays the projection of the cells onto the top two RESET UMAP dimensions. Specifically, UMAP was executed on the top 30 PCs of the BioCarta RESET score matrix. As shown by this projection, the clean clustering by cell type seen in the scRNA-seq UMAP projection is lost though there is still a visible segregation by cell type with some types, e.g., NK cells, retaining a clear separation from the other cells. The blurring of cell type identity is not surprising given the more limited set of genes included in the BioCarta pathways and the fact that RESET scores capture reconstruction error rather than mean abundance.

A important application of the cell-level scores computed by RESET involves the identification and visualization of differential pathway activity. Fig G in S1 Text illustrates such a visualization for the five BioCarta pathways most enriched in each cell type cluster according to the log2 fold-change in the mean RESET score of cells in the cluster relative to cells not in the cluster. Note that differential activity results are insensitive to the per-variable adjustment. These results provide important information regarding the range of pathway activity across all profiled cells. While many of the pathways shown in Fig G in S1 Text align with the expected biology for the associated cell type, e.g., the B cell antigen receptor signaling pathway (as represented by the BCR (B cell receptor) pathway) has elevated scores in B cells and the CTL (cytotoxic lymphocytes) pathway has elevated scores in NK cells, some of the results are unexpected, e.g, CD14+ monocytes have the highest scores for the TCR (T cell receptor) pathway. To correctly interpret the cell-level RESET scores, it is important to remember how the scores are computed mathematically and what that mathematical definition indicates about the structure of the analyzed scRNA-seq data. In particular, RESET scores capture how well a reduced rank representation of a given gene set can reconstruct a reduced rank representation of the entire data set. High cell-level RESET scores indicate that the value of set genes for a given cell can effectively reconstruct the values of all genes for that cell. As illustrated by the simulation studies, this can capture patterns of differential expression, however, it can also identify correlation patterns independent of any mean difference.

Existing single sample gene set scoring methods like VAM, on the other hand, capture differences in mean expression. For such methods, high gene set scores will correspond to cells where mean expression of the genes in the set is elevated relative other cells in the data set. In general, RESET will produce distinct results from mean difference techniques and, for many use cases, performing both types of single sample gene set testing will provide the most comprehensive characterization of the data. Fig 7 illustrates the distinct results generated by RESET and VAM for four of the cells types in the PBMC data (note that the "BIOCARTA-" prefix has been removed to save space). While some pathways appear enriched using either RESET or VAM scores, e.g., BCR (B cell receptor) for B cells, TCYTOTOXIC for CD8 T cells, and CTL (Cytotoxic T Lymphocytes) for NK cells, others are enriched according to just one of the techniques. Biologically relevant pathways that are only enriched according to RESET scores include the ARENRF2 (Oxidative Stress Induced Gene Expression Via Nrf2) [34], AT1R (Angiotensin II mediated activation of JNK Pathway via Pyk2 dependent signaling) [35], and TH1TH2 (Th1/Th2 Differentiation) [36] pathways for Naive CD4 T cells, the NFAT (NFAT and Hypertrophy of the heart) [37] and TALL1 (TACI and BCMA stimulation of B cell immune responses) pathways for B cells, the INFLAM (Cytokines and Inflammatory Response) and GRANULOCYTES (Adhesion and Diapedesis of Granulocytes) pathways for CD8 T cells, and the INFLAM and ASBCELL (Antigen Dependent B Cell Activation) [38] pathways for NK cells. Importantly, these pathways all have a clear biological association with the respective immune cell type (see references in previous sentence) but are not detected by methods like VAM given the very small positive (or negative) differential expression signature.

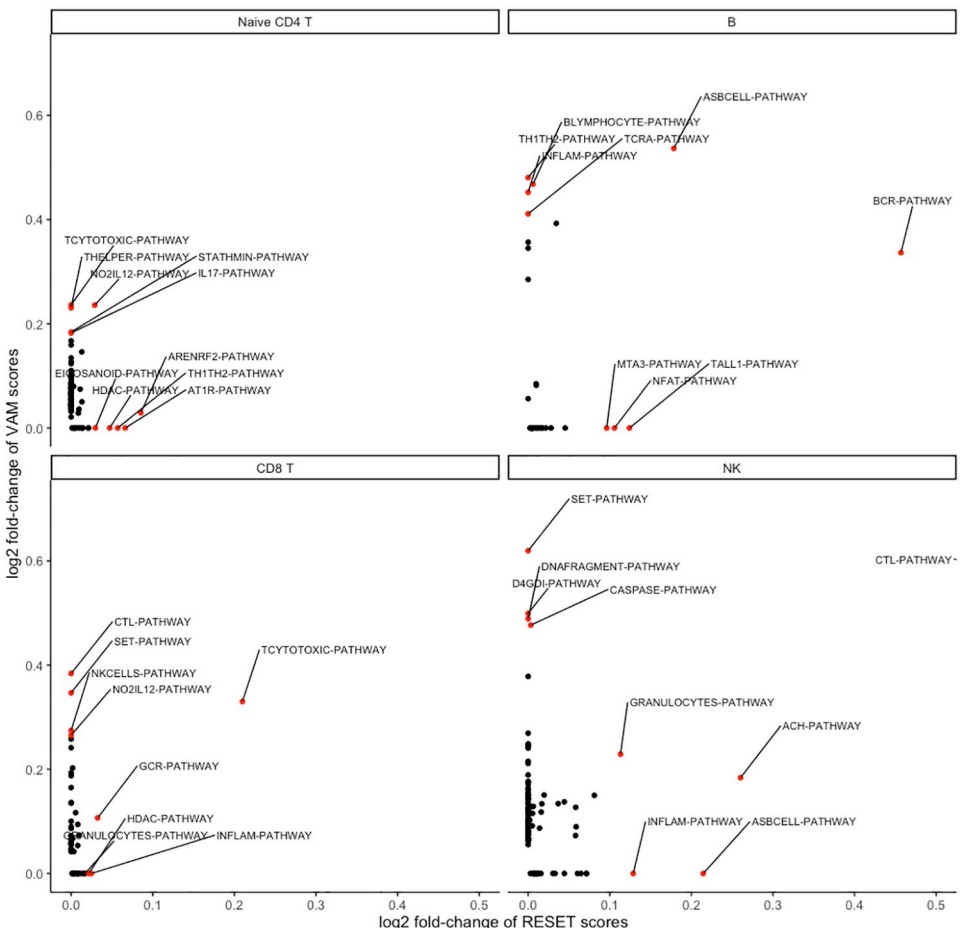

**Fig 7. Visualization of cell type BioCarta pathway enrichment as computed using either VAM or RESET scores on the PBMC scRNA-seq data.**

To explore this behavior, we computed the log2 fold-change (log2fc) in mean normalized gene counts for pathway genes between cells of the target type and all cells. As expected, the pathways detected only by RESET had very small positive DE signatures (or negative DE signatures, which VAM is unable to detect) relative to those found only by VAM. For Naive CD4 T cells, the log2fc in expression for the ARENRF2 and AT1R pathways was 0.17 and 0.12 whereas TCYTOTOXIC and THELPER pathways detected only by VAM had log2fc values of 0.53 and 0.56. Similarly for B cells, the log2fc for the NFAT and TALL1 pathways was -0.22 and -0.09 whereas the BLYMPHOCYTE and TH1TH2 pathways detected only by VAM had log2fc values of 0.98 and 1.15. For CD8 T cells, the INFLAM and GRANULOCYTES pathways had log2fc values of -0.82 and 0.04 whereas the CTL and SET pathways detected only by VAM had log2fc values of 0.46 and 0.96. Lastly for NK cells, the INFLAM and ASBCELL pathways had log2fc values of -1.26 and -1.55 whereas the VAM-only SET and D4DGI pathways had log2fc values of 1.97 and 1.08.

## Human cord blood analysis

To complement the PBMC analysis, we analyzed the 8.6k human cord blood scRNA-seq data set included in the SeuratData R package (see Methods section in S1 Text for details). To

explore the impact of different normalization methods on RESET, we generated results using both Seurat log-normalization and SCTransform. Tables B-E in S1 Text list the top 20 Bio-Carta pathways according to the unadjusted or per-variable adjusted RESET scores for both normalization types. Although the pathway rankings are sensitive to both the per-variable adjustment and the normalization type, in all four cases the overall scores correctly capture the immune cell nature of this data set with a number of pathways appearing in all of the top 20 lists for both the PBMC and cord blood datasets (e.g., MHC, TCRA, CTL, THELPER). Figs J and K in S1 Text illustrate the pathways most significantly enriched in four cell type populations according to either RESET or VAM scores for log-normalization (Fig J in S1 Text) and SCTransform normalization (Fig K in S1 Text). Similar to the PBMC results shown in Fig 7, the enrichment results for the cord blood analysis are consistent with expected biology but with a clear difference in enrichment effect sizes between the two normalization techniques. Although the VAM and RESET results are more similar on the cord blood data than on the PBMC data, there are again a number of pathways detected by only one of the methods. Similar to the PBMC results, these distinct enrichment findings can be explained by the differential abundance signature. For example, the IGF1 (insulin like growth factor 1 signaling) pathway [39] was only found to be enriched in CD4 T cells by RESET for the log-normalized data and had a log2 fold-change in mean expression between CD4 T cells and all other cells of -0.26 whereas the TCYTOTOXIC pathway was only enriched according to VAM scores and had a log2 fold-change of 0.54. For NK cells, the CTLA4 (the co-stimulatory signal during T cell activation) pathway was only enriched according to RESET scores and had a log2 fold-change of 0.2 whereas the CAPASE (caspase cascade in apoptosis) pathway [40] was only enriched according to VAM scores and had a log2 fold-change of 0.98. Results for the other explored cell types had a similar pattern.

## Mouse brain cell analysis

To explore method performance on non-human data measured on a solid tissue, we analyzed the 10x 11.8k mouse brain scRNA-seq data set. For the mouse brain data, we used the SCTransform normalization technique instead of log-normalization and explored a much larger pathway collection (the MSigDB Gene Ontology (GO) biological process (C5.BP) collection). Fig H in S1 Text is a projection of the 9,320 cells remaining after quality control onto the first two UMAP dimensions with labels based on unsupervised clustering results and Fig I in S1 Text is a visualization of pathway enrichment in the six distinct clusters (2, 7, 9, 10, 11, and 12) according to both RESET and VAM scores. In contrast to the PBMC analysis, RESET and VAM scores generated more similar pathway enrichment results for the mouse brain data. Importantly, prioritizing GO terms that appear enriched according to both methods can help identify the likely neuronal cell populations represented by the clusters: gabaeric interneurons for cluster 2, oligodentrocytes for cluster 7, granule cells for cluster 10, vascular cells for cluster 11 and microglial cells for cluster 12.

## Conclusions

Gene set testing is a widely used hypothesis aggregation technique that can improve the power, interpretation and replication of genomic data analyses by focusing on biological pathways instead of individual genes. These benefits are amplified for genomic data generated on individual cells, which has significantly elevated levels of noise and sparsity relative to the output from bulk tissue assays. To address the lack of gene set testing methods optimized for single cell data, we recently developed a new technique for cell-level gene set scoring of single cell transcriptomic data called Variance-adjusted Mahalanobis (VAM). While the VAM technique

offers a significant improvement in terms of computational and classification performance over other single sample methods, it has a number of important limitations. To address these challenges, we developed a new, and analytically novel, single sample method called Reconstruction Set Test (RESET). RESET quantifies gene set importance at both the sample-level and for the entire data based on the ability of genes in each set to reconstruct values for all measured genes. RESET is realized using a computationally efficient randomized reduced rank reconstruction algorithm and can effectively detect patterns of differential abundance and differential correlation for both self-contained and competitive scenarios. An R implementation, which supports integration with the Seurat framework, is available in the RESET package on CRAN. As shown using simulated and real single cell RNA-sequencing data, the RESET method provides superior classification performance at a lower computational cost relative to VAM and other popular single sample gene set testing approaches. Potential future applications/enhancements of the RESET method include gene set optimization (i.e., edit the membership of existing gene sets to maximize the per-variable overall RESET score on a target scRNA-seq dataset), de novo gene set creation (i.e., generate seed gene sets via gene clustering and then optimize the set membership to maximize the per-variable overall RESET score), and exploration of ensemble gene set testing approaches similar to what was shown on the PBMC and mouse brain data using VAM and RESET.

## Supporting information

**S1 Text. Includes supplemental methods and results.**
(PDF)

## Acknowledgments

We would like to acknowledge the supportive environment at the Geisel School of Medicine at Dartmouth where this research was performed.

## Author Contributions

**Conceptualization:** H. Robert Frost.

**Formal analysis:** H. Robert Frost.

**Funding acquisition:** H. Robert Frost.

**Investigation:** H. Robert Frost.

**Methodology:** H. Robert Frost.

**Resources:** H. Robert Frost.

**Software:** H. Robert Frost.

**Validation:** H. Robert Frost.

**Visualization:** H. Robert Frost.

**Writing – original draft:** H. Robert Frost.

**Writing – review & editing:** H. Robert Frost.

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
