## [Decision Letter · Decision Letter 0]

3 Jan 2024

Dear Dr. Frost,

Thank you very much for submitting your manuscript "Reconstruction Set Test (RESET): a computationally efficient method for single sample gene set testing based on randomized reduced rank reconstruction error" for consideration at PLOS Computational Biology.

As with all papers reviewed by the journal, your manuscript was reviewed by members of the editorial board and by several independent reviewers. In light of the reviews (below this email), we would like to invite the resubmission of a significantly-revised version that takes into account the reviewers' comments.

We cannot make any decision about publication until we have seen the revised manuscript and your response to the reviewers' comments. Your revised manuscript is also likely to be sent to reviewers for further evaluation.

Sincerely,

Yang Lu, Ph.D.

Academic Editor

PLOS Computational Biology

Kiran Patil

Section Editor

PLOS Computational Biology

Reviewer's Responses to Questions

**Comments to the Authors:**

Reviewer #1: In this study, the authors developed a randomized numerical linear algebra (RNLA) based computational framework, RESET, for the unsupervised single sample gene set testing. By leveraging the RNLA, the authors aimed to provide a computationally efficient tool. The authors evaluated the performance of REST on both simulated and real single-cell datasets. Although the proposed computational framework is interesting, I have major concerns as stated below:

(1) The main innovation of REST is the use of RNLA for computationally efficient dimensionality reduction. However, this improvement in efficiency over PCA may be marginal. Since the computational cost of PCA is dominated by the small dimension of X (e.g., min (n,p)). Also, n can become large, and the dimensionality of p tends to be ~ 2000 (e.g. for highly variable genes). It seems to me that deterministic PCA will work for most of the task, and that RNLA is not necessary.

(2) The working principles of REST and its ancestor VAM on single sample gene set testing are not explained. The writing style, especially the methods part, is more like a manual. I faced some challenges in understanding the manuscript.

(3) The parameters used in the performance evaluation need to be explained. For example, "classification" of what? Why can "classification" be used for gene set testing? what do "mean inflation", "correlation", "rate" mean?

Minor comments:

(1) I suggest that a diagram of the RESET workflow be provided. Some details on the RNLA used would also be helpful.

(2) How does RESET deal with the batch effect when samples are collected and sequenced in different batches?

Reviewer #2: The paper proposed a new method called Reconstruction Set Test (RESET) for single sample gene set testing. RESET quantifies the importance of gene sets based on their ability to reconstruct values for all measured genes. It uses randomized reduced rank reconstruction algorithm to detect patterns of differential abundance and differential correlation for both self-contained and competitive scenarios. The method generates both overall and sample-level scores for evaluated gene sets.

However, there are some issues, which must be solved before it is considered for publication.

1.Why the RNLA, more than PCA and Z-score[1], is effective in detecting patterns of differential abundance and differential correlation in independent and competing scenes(In section 2.1).

2.More experiments should be offered to demonstrate the validity and applicability of the proposed method. For example, PLAGE and PAGODA are added to the comparison methods and explain why the RESET can detect competitive scenarios where the measured values of set genes differ from non-set genes in the same sample.

3. The simulation results should be explained in detail, such as why the higher inter-gene correlation indicates the better the RESET results？

Minor: The Introduction section does not provide sufficient background information or context to help readers understand the significance of your research. We suggest that you consider adding more detail, such as what are the competitive and self-contained features.

1. Tabaka M, Gould J, Regev A. scSVA: an interactive tool for big data visualization and exploration in single-cell omics, bioRxiv 2019:512582.

**Have the authors made all data and (if applicable) computational code underlying the findings in their manuscript fully available?**

Reviewer #1: Yes

Reviewer #2: None

PLOS authors have the option to publish the peer review history of their article (what does this mean?). If published, this will include your full peer review and any attached files.

Reviewer #1: No

Reviewer #2: No
---

## [Decision Letter · Decision Letter 1]

3 Apr 2024

Dear Dr. Frost,

Thank you very much for submitting your manuscript "Reconstruction Set Test (RESET): a computationally efficient method for single sample gene set testing based on randomized reduced rank reconstruction error" for consideration at PLOS Computational Biology. As with all papers reviewed by the journal, your manuscript was reviewed by members of the editorial board and by several independent reviewers. The reviewers appreciated the attention to an important topic. Based on the reviews, we are likely to accept this manuscript for publication, providing that you modify the manuscript according to the review recommendations.

Sincerely,

Yang Lu, Ph.D.

Academic Editor

PLOS Computational Biology

Kiran Patil

Section Editor

PLOS Computational Biology

Reviewer's Responses to Questions

**Comments to the Authors:**

Reviewer #1: My concerns have been addressed.

Reviewer #2: The authors have revised the paper carefully according to the comments. I think the paper can be accepted in the current version.

Reviewer #3: The author(s) developed a new gene set testing method RESET, which quantifies gene set importance for transcriptomics datasets with multiple samples/conditions. The implementation of RESET was introduced clearly by first setting the basic RESET method of constructing the full expression matrix with a subset of genes in a gene set, and then adding in the rank reduction through PCA component, and finally adding randomized numerical linear algebra for increased computational efficiency. The revised manuscript provides sufficient motivation for RESET in the introduction, and improved explanation of the improvements of RESET over existing methods. The method will be useful for interpretation of transcriptomics datasets.

I have only one major comment and a few minor comments/questions.

The major comments relates to the focus on single cell RNA-seq applications.

-It would help to clarify more up front that for single cell datasets the scores are calculated on each cell. The term sample could mean the individual replicates, and indicate that the RESET scores are calculated on pseudobulk samples.

-Since RESET calculates cell level scores, a natural comparison would be AUCell from the SCENIC pipeline. Does RESET scores correlate with AUCell scores well? Are RESET scores more informative than AUCell scores?

-Is there intuition on how deeply sequenced each cell needs to be (or the sparsity level of the dataset) for RESET to perform effectively?

-For the real single cell datasets, if the cell by gene set scores matrix is projected and visualized, or clustered, do cells of the same type still cluster together?

-I think the possible ways to utilize cell level RESET scores for interpretation of single cell datasets should be mentioned in the methods section.

The minor comments are as follow:

-Are all genes in X required to be in at least one gene set?

-A few terms could be better introduced in the text:

--When the concept of 'Self-contained vs competitive' was first introduced in section1.1, the definition was a more mathematical one, while in section1.2 the 'Self-contained methods' is described in more layman terms 'generate scores using only the data for genes in the set'. I think it'd help to make a clear connection between these descriptions or consolidate.

--The term 'classification performance' is still somewhat confusing. I think more clearly defining the classification task, if there is one, would help.

--The term 'differential correlation' was mentioned early on but not defined until the results section (where correlation was explained but still not clearly defined).

-Is novel gene set discovery a possibility within this framework?

**Have the authors made all data and (if applicable) computational code underlying the findings in their manuscript fully available?**

Reviewer #1: Yes

Reviewer #2: None

Reviewer #3: Yes

PLOS authors have the option to publish the peer review history of their article (what does this mean?). If published, this will include your full peer review and any attached files.

Reviewer #1: No

Reviewer #2: **Yes: **Ying Wang

Reviewer #3: No

Figure Files:

Data Requirements:

Reproducibility:

References:

---

## [Decision Letter · Decision Letter 2]

17 Apr 2024

Dear Dr. Frost,

We are pleased to inform you that your manuscript 'Reconstruction Set Test (RESET): a computationally efficient method for single sample gene set testing based on randomized reduced rank reconstruction error' has been provisionally accepted for publication in PLOS Computational Biology.

Best regards,

Yang Lu, Ph.D.

Academic Editor

PLOS Computational Biology

Kiran Patil

Section Editor

PLOS Computational Biology

Reviewer's Responses to Questions

**Comments to the Authors:**

Reviewer #3: The author has addressed my questions. I have only one remaining comment regarding 3.1.2. I agree with the author that comparing to GSVA and ssGSEA is sufficient. I think what would be helpful though is a small section showing the concordance of results between methods (i.e. across single samples/cells, are the scores from different methods for the same pathway well correlated? This would be similar to Fig. 7 but at single sample/cell level, and scores instead of FC). And if there is a lack of concordance, what are the speculations that would explain the differences?

**Have the authors made all data and (if applicable) computational code underlying the findings in their manuscript fully available?**

Reviewer #3: Yes

PLOS authors have the option to publish the peer review history of their article (what does this mean?). If published, this will include your full peer review and any attached files.

Reviewer #3: No

---

## [Editor Report · Acceptance letter]

25 Apr 2024

PCOMPBIOL-D-23-01563R2 

Reconstruction Set Test (RESET): a computationally efficient method for single sample gene set testing based on randomized reduced rank reconstruction error

Dear Dr Frost,

I am pleased to inform you that your manuscript has been formally accepted for publication in PLOS Computational Biology. Your manuscript is now with our production department and you will be notified of the publication date in due course.

With kind regards,

Anita Estes
